# The influence of trace metal supplementation on the presence of ceftriaxone resistance in *Enterobacteriaceae* in the gastrointestinal tract of dairy cattle

Charles-Antoine Martineau,[1,2] Mélissa Duplessis,[1] Jennifer Ronholm,[3,4] Renée M. Petri[1]

**ABSTRACT** The addition of trace minerals into the diet of lactating cows frequently exceeds national recommendations for industry practices. However, the presence of certain heavy metals, such as zinc and copper, has been shown to exert selection pressure on the gut microbiota, favoring metal resistance and potential co-selection for antimicrobial resistance. To determine whether oversupplementation of dietary zinc alters the gut microbiota of dairy cattle, a cross-over design was used to feed either recommended or surplus levels of dietary zinc (0.89×; high mineral diet) compared to the recommended levels (control diet). Rumen, duodenum, and fecal samples were collected to analyze the 16S rRNA microbial community for diversity and relative abundance, with a greater focus on the *Enterobacteriaceae* family, while mixed enriched gut content samples were cultured to determine the presence of zinc, copper, and ceftriaxone resistances in gram-negative bacteria. Alpha-diversity analysis showed a decrease in richness and evenness (Simpson index) when cows were in the HIGH treatment ($P = 0.0464$) and a tendency to decrease ($P = 0.0592$) diversity according to the Shannon index. Despite alpha-diversity differences, *Enterobacteriaceae* abundance showed no difference between treatments. For culturing, a tendency ($P = 0.0956$) for decreased fecal resistance to zinc on MacConkey mixed enriched isolates was observed for the HIGH group. This study showed that there were differences between niches but no significant increase in resistance in response to zinc, copper, and ceftriaxone in the enriched *Enterobacteriaceae* populations from the rumen, duodenum, and fecal niches and that zinc oversupplementation had minimal impact on gut microbiota communities.

**IMPORTANCE** The addition of trace minerals into the diet of lactating cows, at levels exceeding national recommendations, is a common industry practice. However, there are new concerns as the presence of certain heavy metals, such as zinc, has been shown to exert selection pressure on the gut microbiota, favoring metal resistance and potential co-selection for antimicrobial resistance. We evaluated how the addition of zinc in the diet of lactating cows affects the bacterial community's relative abundance and diversity, with a focus on the *Enterobacteriaceae* family throughout the gastrointestinal tract, due to their importance for human health. Using samples from the rumen, duodenum, and feces, we cultivated gram-negative bacteria from enriched samples in the presence of zinc, copper, and ceftriaxone resistances to confirm phenotype resistances. This study contributes to our understanding of how dairy diets oversupplemented with minerals may alter the microbial community of the animal and could contribute to the dissemination of antibiotic resistance.

**KEYWORDS** 16S rRNA, AMR, rumen, duodenum, ceftriaxone, dairy, resistance, zinc

**Peer Reviewers** Alexandra Calle, Texas Tech University, Amarillo, Texas, USA; Hassan M. Al-Tameemi, Basrah University, Basrah, Iraq

Address correspondence to Renée M. Petri, renee.petri@canada.ca.

The authors declare no conflict of interest.

Modern lactating cows are highly efficient, regularly reaching 20,000 kg of milk per lactation (1). This high level of production frequently exceeds the animal's energy intake capacity but can also result in mineral deficiencies, leading to several illnesses including milk fever, lameness, depressed immunity leading to mastitis, and even reduced fertility (2–4). To palliate potential illnesses, lactating cows often receive an oversupplementation of minerals in the lactation diet (5) to support immunity, oxidative metabolism, and prevent illness (6). However, current nutritional recommendations for trace mineral supplementation do not take into consideration the impact of oversupplementation on the gastrointestinal ecosystem, more specifically the general health and growth of the gut microbiome. This is an area of concern as some trace minerals are heavy metals, including copper and zinc, which are known to exhibit antimicrobial effects (7) that can contribute to the dissemination of antimicrobial resistance (AMR) by enriching resistance gene determinants (metal resistance genes) via co-selection mechanisms (8, 9), even in livestock animals (10). Despite extensive research, the duration of exposure necessary to induce metal or co-selection of antibiotic resistance in the gut microbiota of livestock remains poorly understood. Given that dairy cattle are fed mineral-rich diets for substantially longer periods compared to other livestock species, it is crucial to elucidate the impact of these feeding practices on gut microbial dynamics. Antimicrobial resistance gene dissemination is especially important for One Health when discussing dairy gut commensal microbiome populations, such as the *Enterobacteriaceae* family, including *Escherichia coli*, which are pathogenic to humans and known for carrying antibiotic resistance (11, 12). Furthermore, *Enterobacteriaceae* are gram-negative bacteria that are commonly used as viable indicators for fecal contamination and are identified as priority pathogens for public health (13). They are also known to be hosts of clinically important antibiotic resistance genes, as well as being identified as having metal resistance genes (14), making them excellent candidates for understanding the transmission of AMR.

The emergence of such resistant bacteria often occurs in food animals when antimicrobials are used (15). One commonly used drug in dairy cows is ceftiofur, a third-generation cephalosporin that is labeled for veterinary use in food and companion animals to treat a variety of gram-negative bacterial infections (16) because of its low risk for milk residues as well as its perceived efficacy (17). Ceftriaxone is a third-generation cephalosporin used in human medicine similar to ceftiofur, as well as in the treatment of salmonellosis in children and adults; however, resistance to ceftriaxone and other third-generation cephalosporins in *Salmonella* spp. seems to be increasing (18). Comparatively, in dairy cattle, previous research has shown a reduced susceptibility to ceftriaxone in the *E. coli* population from feces (16). Therefore, based on the usage of metals in dairy diets and the prevalence of resistant populations in dairy cattle feces, an increased understanding of how the presence of heavy metals in the diet could impact resistance to clinically relevant antibiotics in humans, such as ceftriaxone, is critical in assessing the impacts of dairy on the dissemination of AMR.

Diet is a major influencing factor on the gut microbiome (19), and while the impact of diet on rumen and fecal microbiomes is relatively well understood due to ease of sampling, there is significantly less known about the microbiome of the small intestine digesta and its role in the passage of microbiota (20). To date, many studies have characterized the impact of age (21) or genetics on the gastrointestinal microbiota (22, 23) or have assessed the presence of predicted genes in the microbiota population related to metabolism (20). However, there is limited information about the characterization of gastrointestinal niches related to the feeding of heavy metals. While the advancement of gut microbial knowledge through sequencing platforms has increased our understanding of the presence of resistance genes, relatively less is known about the relationship between diet and phenotypic antibiotic resistance found through culturing.

Therefore, in this study, we assessed the microbiota from various gastrointestinal niches, including the rumen, duodenum, and feces, to determine the potential emergence of phenotypic resistance to zinc, as well as co-resistance to copper and

ceftriaxone based on dietary mineral supplementation commonly found in commercial dairy farms. We hypothesized that the oversupplementation of zinc in the diet would alter the microbial community in different parts of the gastrointestinal tract and that there would be an increased phenotypic resistance to other heavy metals (copper) and antibiotics, such as ceftriaxone, by association.

## RESULTS

### pH

Rumen spot sample pH varied over the day according to the time of feeding (Fig. 1), whereas duodenal pH remained constant. The average daily pH was significantly different between niches ($P = 0.001$). The feces had the highest value ($7.08 \pm 0.23$) followed by the rumen ($6.33 \pm 0.40$), and the lowest was the duodenum ($2.56 \pm 0.30$). No difference was observed between the control diet (CON) and high mineral (HIGH) diet for the feces or rumen niches, but there was a tendency toward a lower pH in the HIGH treatment group for the duodenum ($P = 0.07$; 2.64 vs 2.47: CON vs HIGH, respectively).

### Microbial diversity

Analysis of the alpha diversity for the duodenum showed a higher diversity in the CON ($0.99 \pm 0.01$) diet compared to HIGH ($0.97 \pm 0.03$) diet ($P = 0.0464$) when looking at the Simpson indices and tended to increase ($P = 0.0592$) for the Shannon index ($4.99 \pm 0.30$ and $4.67 \pm 0.51$) between the CON and HIGH diets, respectively. Similarly, in the rumen, the Simpson index tended toward increased diversity in the CON compared to the HIGH diets ($P = 0.0851$). No significant differences were seen between diets for the rumen ($P = 0.1261$) and feces ($P = 0.4806$) for the Shannon index, nor the Simpson index in the feces ($P = 0.4495$; Fig. 2).

Analysis of beta diversity using the Bray-Curtis dissimilarities showed clustering between the rumen and duodenum communities compared to the feces ($P = 0.001$), but no significant differences were observed between CON and HIGH groups (Fig. 3).

### Relative abundance

Relative abundances were compared for the phylum level of taxonomy between the different niches (Fig. 4). The analysis for some phylum (*Actinobacteriota*, *Fibrobacterota*, *Fusobacteriota*, *Proteobacteria*, *Spirochaetota*, and *Verrucomicrobiota*) was not validated by the normality of the residuals, and the presence of *Fusobacteriota* was only detected in one animal.

There was a higher abundance of *Bacteroidota* in the rumen compared to the duodenum ($P = 0.0002$) and the feces ($P < 0.0001$), which decreased with passage through the gastrointestinal tract. The relative abundance of *Firmicutes* increased with passage through the gastrointestinal tract, from lowest in the rumen, increasing in the duodenum ($P = 0.0295$), and higher still in abundance in the feces *($P < 0.0001$)* compared with the rumen and duodenum. *Fibrobacterota* and *Verrucomicrobiota* were most abundant in the duodenum compared to the rumen ($P < 0.02$) and the feces ($P = 0.009$). *Spirochaetota* had a higher abundance in the rumen compared to the duodenum ($P = 0.0420$), and *Actinobacteriota* abundance was highest in the feces compared to the rumen ($P = 0.0015$) and the duodenum ($P = 0.0024$). *Proteobacteria* was increased in abundance in the duodenum compared to the rumen ($P = 0.0289$). Therefore, significant variations in members of the *Proteobacteria* were further analyzed between niches as no differences between treatments were observed ($P = 0.40$).

Statistical assessment of the identified members of the *Enterobacterales* order showed a significant interaction between niche and treatment (Table 1). When assessing the niche differentiation, the relative abundance of the order *Enterobacterales* was the highest for the duodenum compared to the other niches ($P < 0.0001$), and there was an increasing tendency for the rumen compared to feces ($P = 0.0756$; Fig. 5A). The significant interaction between niche and treatment was also seen for the *Succinivibrionaceae* family

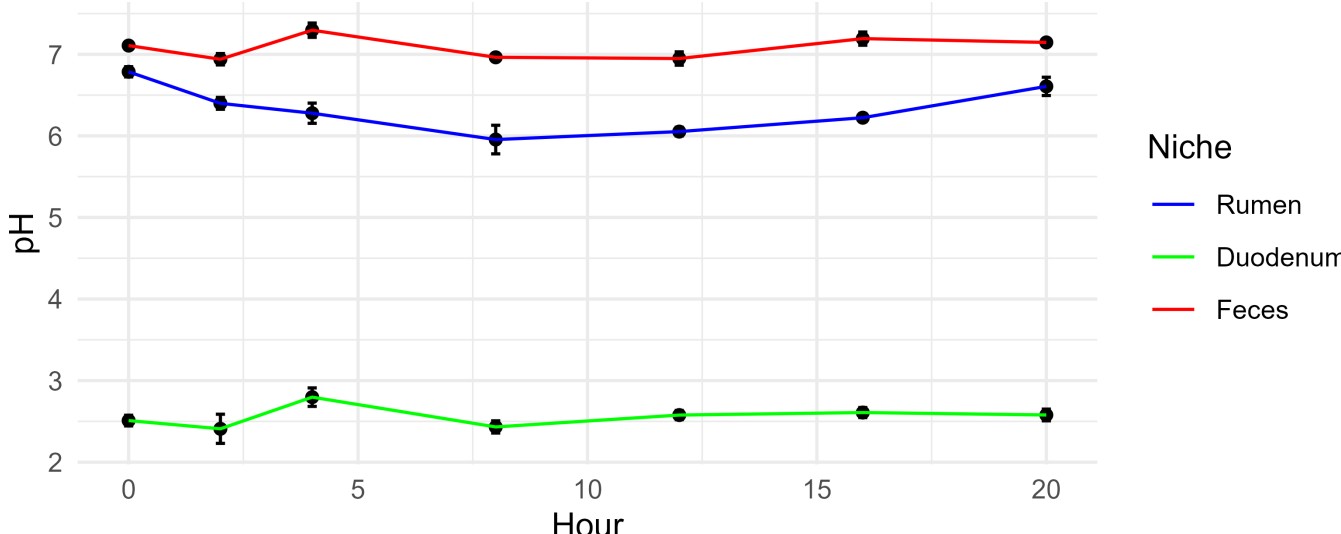

**FIG 1** Daily variation in spot-sampled pH for each niche (rumen, duodenum, and feces) relative to the number of hours post-feeding.

and the genera *Succinivibrio* and *Succinivibrionaceae* UCG002. In the rumen and the duodenum, the *Enterobacteriaceae* family showed a higher relative abundance than the feces ($P < 0.0001$; Fig. 5B). Comparison of other members of the *Enterobacteriaceae* family, *Escherichia-Shigella*, and the *Succinivibrionaceae* family, *Ruminobacter*, and *Succinimonas* spp., showed that they did not have interaction effects but were significantly impacted by niche (Table 1; Fig. 5C). Specifically, *Ruminobacter* and *Succinimonas* were highest in relative abundance in the duodenum, whereas *Escherichia-Shigella* was found exclusively in the feces.

## Culturing and phenotypic resistance

The culturing of mixed enriched isolates on selective media for gram-negative bacteria showed no significant effect between dietary treatments and no growth on combinations between metals and/or ceftriaxone. The only notable effect was a tendency ($P = 0.0956$) of decreased fecal zinc resistance for the group receiving HIGH dietary treatment (Fig. 6). Samples from the duodenum showed weak growth on MacConkey media, only 70.8% with $8 \pm 26$ UFC were collected as mixed enriched isolate (MEI), probably due to the very low pH of that niche ($2.56 \pm 0.30$).

## DISCUSSION

In this study, we assessed the microbiota from the rumen, the duodenum, and the feces niches to determine the potential emergence and transmission of resistance to copper, zinc, and ceftriaxone through the gastrointestinal tract based on dietary oversupplementation of trace minerals. We hypothesized that oversupplementation of zinc in diet would alter the microbial community in different parts of the gastrointestinal tract and that there would be an increased phenotypic resistance to copper, zinc, and ceftriaxone, as these combinations have been demonstrated to happen in the environment (8, 24).

Statistical variation was seen between the HIGH and CON groups for bacterial diversity in the duodenum for the Simpson index. The Simpson's index ($D$) takes into account the summation of individual groups divided by the summation of all groups in the community, so when one group dominates in the community, it results in a higher value that represents a lower evenness. Because we have Simpson's index at its complement ($1-D$), we can conclude that increasing dietary minerals augments the dominance of certain bacterial groups in the duodenum, as the Simpson's index was higher for the CON compared to the HIGH diet. This would indicate that the duodenum bacterial diversity was more susceptible to some trace mineral addition than the

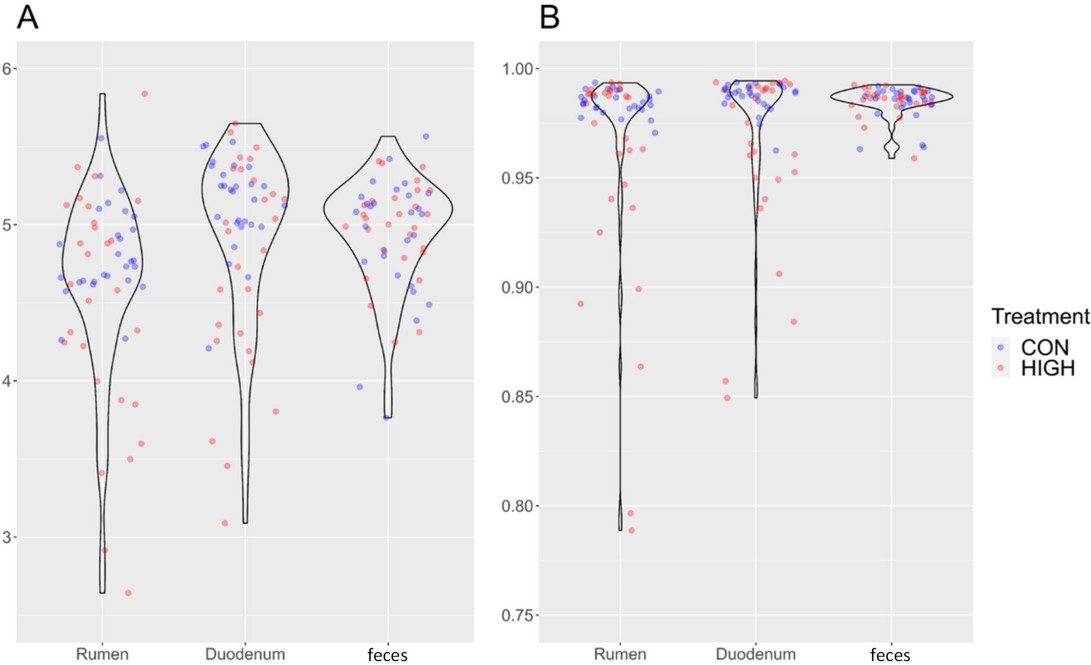

**FIG 2** (A) Shannon alpha diversity indexes. (B) Inverse Simpson alpha diversity indexes sorted by niche (rumen, duodenum, and feces) and for the control diet and high mineral diet.

rumen or the feces. This variation in the duodenum is possibly due to lower pH found after exiting the abomasum altering the solubility of the metals in different niches of the gut. Most minerals adopt a positive charge in acidic conditions, augmenting their effect on the negatively charged bacterial outer membrane (25). This effect could disrupt bacterial growth and activity (8), which is further supported by the lack of microbial growth seen in the duodenal samples when cultured on enriched media. Possible additional reasons for the difficulty in growing isolates from duodenal samples include the choice of selective media for targeting gram-negative bacteria, as well as the limitation of growing only those gastrointestinal bacteria that can be cultivated under aerobic conditions. However, due to the low numbers of individuals enrolled in this study, it is important not to overinterpret the results, especially seeing as there is limited literature for comparison. In the current study, duodenal pH was on average 2.56 ± 0.30, compared to the rumen (6.33 ± 0.40) and the feces (7.08 ± 0.23). In contrast, Mao et al. (20) looked at the microbial differences in the gastrointestinal gut of dairy cattle and showed that duodenum pH was 3× higher (pH = 7.8) than what was found in this study despite the higher starch content fed to animals. One possible explanation for these variations may be a result of sampling style. In the current study, animals were fitted with duodenal cannulas, and pH samples were taken repeatedly throughout a 24 h period compared to the previously mentioned literature where samples were obtained from euthanized animals. This is supported by older research with sampling from live animals, which reported a pH range between 2.41 and 2.32 when starch content was augmented, respectively (26). As a result, it is clear that an accurate depiction of the small intestinal ecosystem is not easily obtainable using terminal experiments, and therefore, interpretation of such results should be done with acknowledgment of the possible biases being introduced through the sampling procedure.

In support of the changes in diversity, there were significant increases in the relative abundance of *Proteobacteria*, *Fibrobacterota*, and *Verrucomicrobiota* in the duodenum compared to both the rumen and feces. However, these increases were not identifiable to a specific family or genus, nor were they associated with an effect of treatment. Using 16S analysis, the relative abundances of *Enterobacteriaceae*-associated genera did

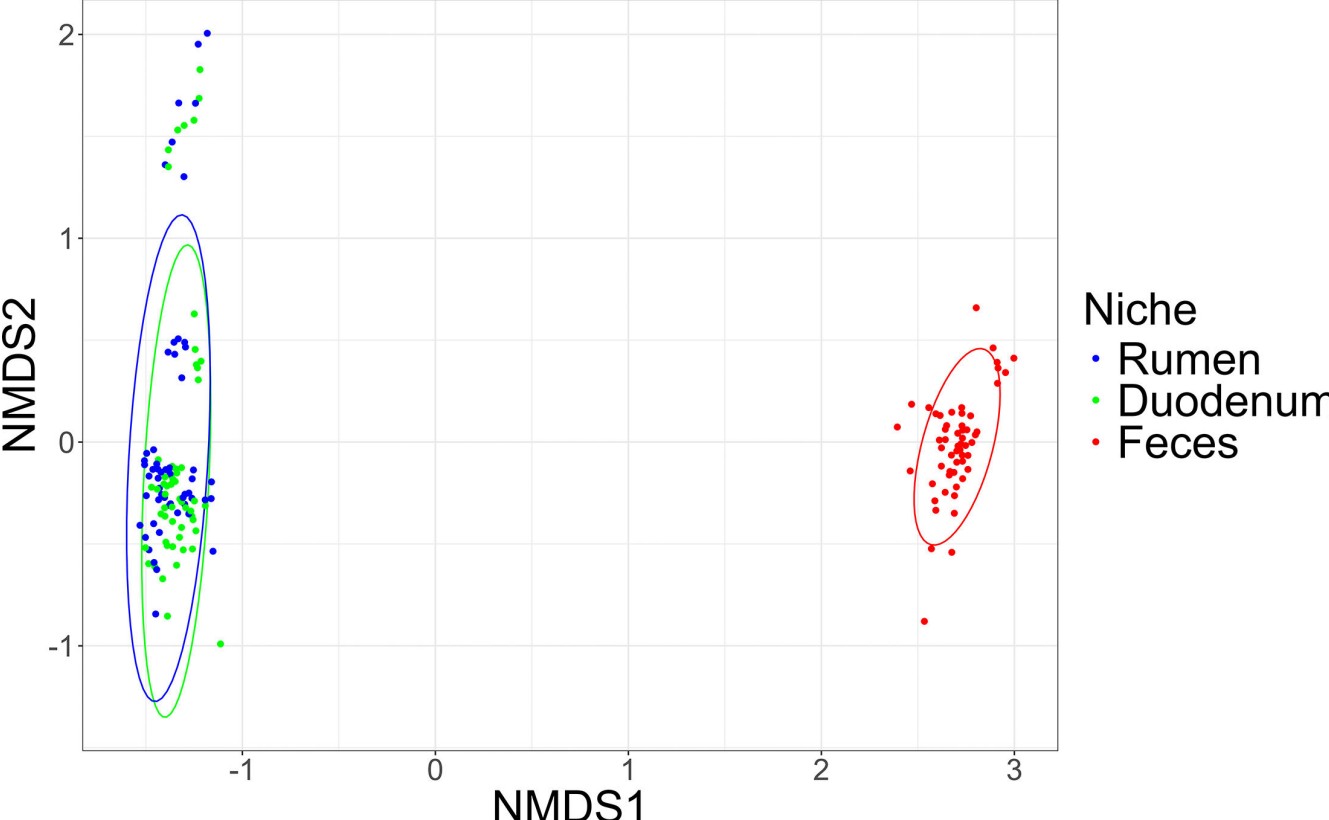

**FIG 3** Bray-Curtis nonmetric multidimensional scaling ordination between gastrointestinal niches (rumen, duodenum, and feces) bacterial communities with a significant difference between niches determined with permutational analysis of variance ($P = 0.001$).

not vary because of the oversupplementation of minerals. Therefore, further analysis is required to verify the hypothesis of pH impacting metal solubility in the gastrointestinal tract using a method that more accurately identifies bacterial populations. It would also be preferable to look at the comparisons between duodenal content and mucosal populations, as mucosal populations likely have less variability and pH sensitivity due to the protective mucosal layer, as seen in other species (27).

While variation along the gastrointestinal tract is expected due to variations in physiological and nutritional parameters, this study showed increased similarity in beta diversity between the rumen and duodenum relative to the feces. This is interesting as there was a clear variation in the pH between the rumen and the duodenum, while the fecal pH in this experiment was closer in value to the rumen. This similarity could be explained by the spatial closeness of rumen and duodenum. However, the inability to grow microbes from duodenal contents would imply a bias in the 16S data that could be associated with the extraction of microbial DNA from dead or ruptured microbial cells. These types of biases are critical in understanding the gastrointestinal niche as a whole, rather than being separated into individual niches. To determine if genera associated with potential human pathogens within the *Enterobacteriaceae* taxa were possibly dominant between niches and to try to make links between the emergence of resistance phenotypes, we compared the relative abundances at the level of the order *Enterobacterales*. Despite the limit of precision due to the sequencing of the 16S ribosomal DNA region, we were able to determine some differences. Although the general relative abundance was more similar between the rumen and the duodenum, when focusing on *Enterobacterales*, the duodenum shows the greater relative abundance of all niches. However, this difference is explained by the greater amount of *Succinivibrionaceae*, which is no longer considered *Enterobacterales* (28). When focusing only on

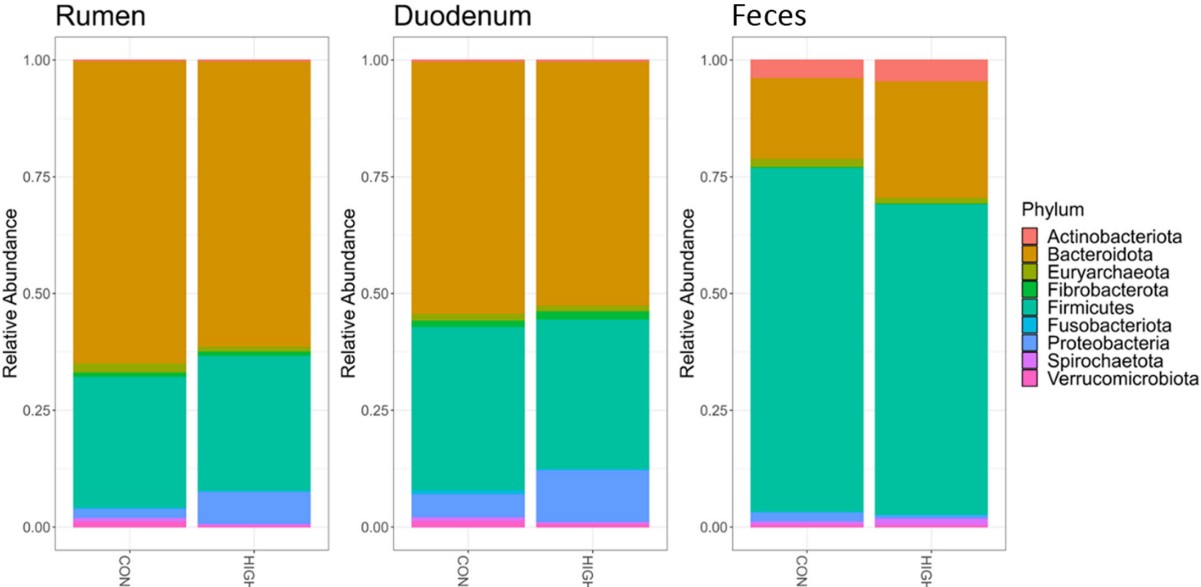

**FIG 4** Relative abundance for the nine principal phyla (*Actinobacteriota*, *Bacteroidota*, *Euryarchaeota*, *Fibrobacterota*, *Firmicutes*, *Fusobacteriota*, *Proteobacteria*, *Spirochaetota*, and *Verrucomicrobiota*) separated by niche (rumen, duodenum, and feces) for the control diet and high mineral diet.

the *Enterobacteriaceae* family, two genera were determined to be significantly different between niches: *Raoultella* spp. was exclusively found in the rumen, and *Escherichia-Shigella* spp. was exclusively found in the feces. *Raoultella* spp. are natural inhabitants of soil, water, and the gastrointestinal tract but can also be opportunistic pathogen for humans with an impaired immune system (29). Moreover, Koc et al. (30) found that one species of *Raoultella* isolated from water was resistant to 15 drugs and 11 heavy metals. This is particularly alarming since they could serve as vessels to transmit resistance genes to more virulent microorganisms, such as *Escherichia* spp. and *Shigella* spp., both of which are particularly important when addressing human health. Furthermore, these two genera are also occasionally found with genes implicated in defense against multiple antibiotics or metals, and when exposed to some heavy metals, it can exacerbate their capability to conjugate resistance genes (31–33).

To better understand the potential resistance phenotypes, MEIs of gram-negative enteric bacteria from all niches were plated on selective media with copper, zinc, and

**TABLE 1** Relative abundance of members of the phylum *Proteobacteria* and order *Enterobacterales*

| | CON | | | HIGH | | | | | | |
| --- | --- | --- | --- | --- | --- | --- | --- | --- | --- | --- |
| | Rumen | Duodenum | Feces | Rumen | Duodenum | Feces | SEM | Niche | Treatment | Niche × treatment |
| Order | | | | | | | | | | |
| *Enterobacterales* | 10.01 | 24.54 | 11.84 | 21.49 | 35.37 | 7.95 | 6.621 | <0.0001 | 0.537 | 0.006 |
| Family | | | | | | | | | | |
| *Enterobacteriaceae* | 0.00 | 0.00 | 0.73 | 0.06 | 0.00 | 0.82 | 0.184 | <0.0001 | 0.824 | 0.948 |
| *Succinivibrionaceae* | 10.00 | 24.53 | 11.73 | 21.48 | 35.37 | 7.62 | 6.565 | <0.0001 | 0.538 | 0.005 |
| Genus | | | | | | | | | | |
| *Anaerobiospirillum* | 0.06 | 0.08 | 0.00 | 0.00 | 0.18 | 0.00 | 0.083 | 0.118 | 0.883 | 0.450 |
| *Escherichia-Shigella* | 0.00 | 0.00 | 0.73 | 0.00 | 0.00 | 0.82 | 0.174 | <0.0001 | 0.887 | 0.933 |
| *Raoultella* | 0.00 | 0.00 | 0.00 | 0.06 | 0.00 | 0.00 | 0.025 | 0.370 | 0.391 | 0.370 |
| *Ruminobacter* | 4.65 | 15.95 | 2.82 | 5.65 | 14.98 | 2.20 | 2.226 | <0.0001 | 0.949 | 0.669 |
| *Succinimonas* | 0.00 | 0.57 | 0.00 | 0.48 | 1.34 | 0.00 | 0.373 | <0.0001 | 0.431 | 0.218 |
| *Succinivibrio* | 1.05 | 3.21 | 10.07 | 2.68 | 4.15 | 6.66 | 2.012 | <0.0001 | 0.921 | 0.033 |
| *Succinivibrionaceae* UCG001 | 0.00 | 0.00 | 0.00 | 0.20 | 0.36 | 0.00 | 0.152 | 0.129 | 0.391 | 0.129 |
| *Succinivibrionaceae* UCG002 | 6.74 | 12.14 | 0.00 | 18.36 | 26.11 | 0.00 | 6.199 | <0.0001 | 0.381 | 0.008 |

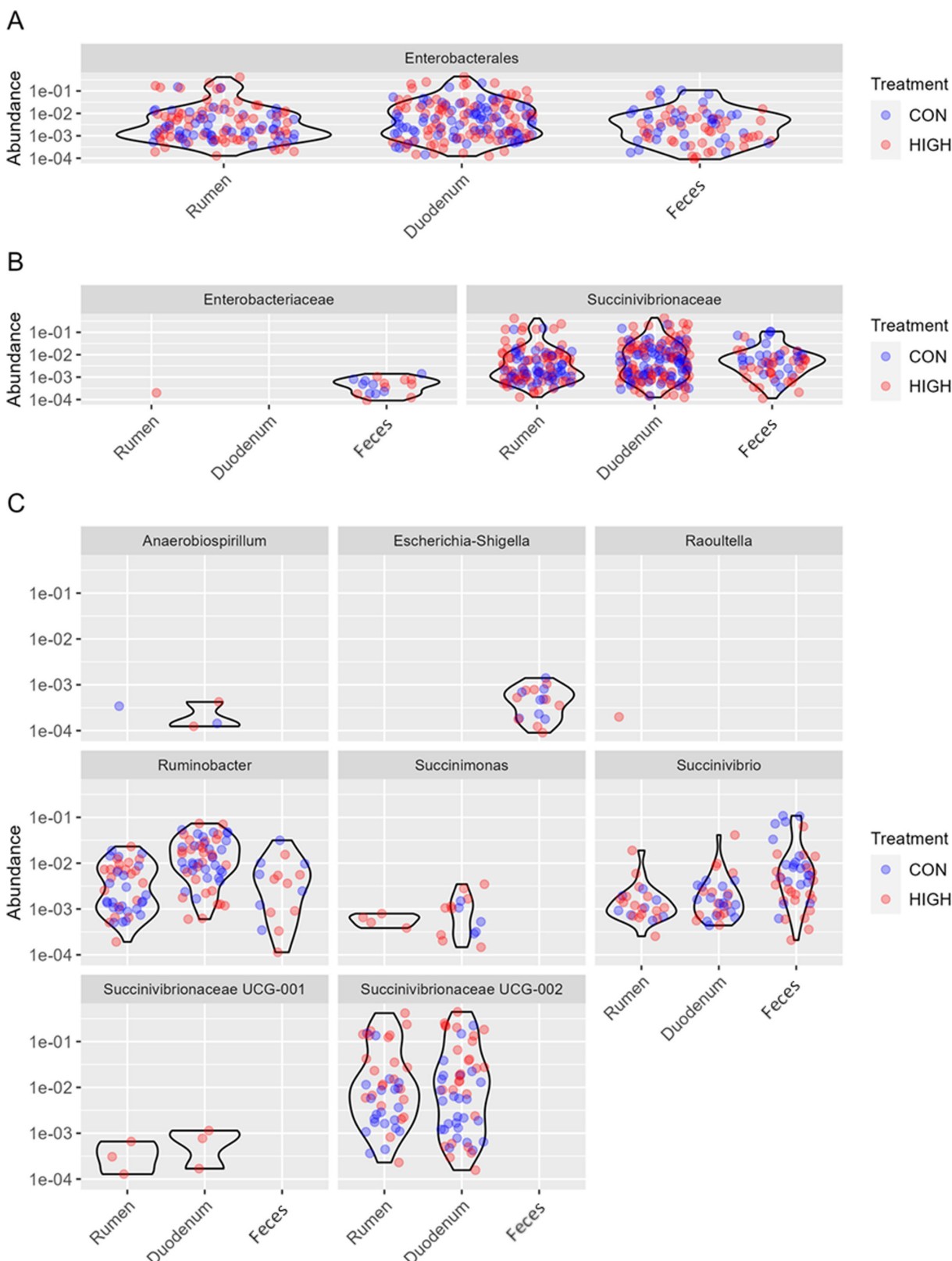

**FIG 5** (A) Relative abundance for members of the Enterobacteriaceae order. (B) Relative abundance for the family members of the Enterobacteriaceae order. (C) Relative abundance for the genus members of the Enterobacteriaceae order sorted by niche (rumen, duodenum, and feces) for the control diet and high mineral diet.

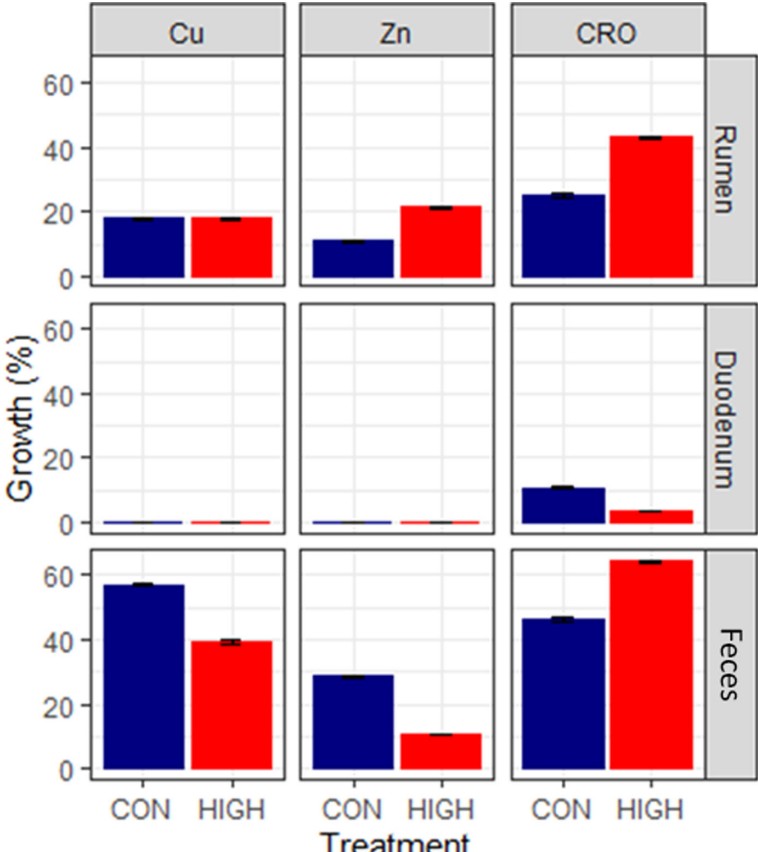

**FIG 6** Growth percentage (%) from presence/absence counts of mixed enriched isolates on MacConkey for each selective media for copper (Cu; blue), zinc (Zn; red), and ceftriaxone (CRO; green) displayed by niches (rumen, duodenum, and feces) and treatment (control: CON; high mineral: HIGH), with error bars indicating standard deviation.

ceftriaxone. This selection process was chosen in order to potentially identify changes within Enterobacteriaceae family members that are commensal to the gut but also of interest to public health, such as *E. coli*, *Enterobacteria*, *Klebsiella*, *Salmonella*, *Proteus*, *Yersinia*, and *Pseudomonas*. There was no significant effect of the diet on the resistance phenotype. The only notable difference was a tendency toward reduced resistance in feces for HIGH treatment samples plated on zinc media. This result was surprising as metal concentrations, such as those used in our culturing procedure, that are below the minimal inhibitory concentration (MIC) are known to generate resistance (34). One potential explanation could be the dynamics of the rumen and the solubility of the metals in the feed altering the capacity for supplemented metals to initiate any resistance phenotype in bacteria. In general, papers that demonstrate a strong correlation between metal exposure and resistance come from environmental sampling where the microorganisms are in contact with high levels of metals for an extended period of time (8). While dietary exposure was sufficient for altering the access of gut microbiota to available nutrients, it is possible that the form of copper and zinc was not accessible to the entire gastrointestinal ecosystem. It is also difficult to note any effect of the treatments between the niches due to the strong difference in pH and variations in enrichment techniques used for the collection of gram-negative bacteria MEI from the duodenum. Variations in the methodology between targeted gene sequencing and selective culturing make correlations between the data sets difficult. However, the trend was similar in that higher amounts of resistant isolates, as well as predicted resistance genes, were found in the feces. While these genes could not be connected to 16S sequences, the data indicate similar trends.

**TABLE 2** Total mixed ration ingredient composition for the basal diet for control (CON) and excess (HIGH) mineral supplementation

| Ingredient (% of DM) | CON | HIGH | SD[a] |
|---|---|---|---|
| Corn silage | 32.94 | 32.92 | 1.13 |
| Grass silage | 30.02 | 30.00 | 1.50 |
| Dry hay | 1.72 | 1.72 | 0.02 |
| Cracked corn | 19.05 | 19.03 | 0.40 |
| Soybean meal | 5.07 | 5.07 | 0.11 |
| Supplement[b] | 8.69 | 8.68 | 0.18 |
| Prolact[c] | 0.74 | 0.74 | 0.02 |
| Calcium | 0.86 | 0.86 | 0.02 |
| Minerals[d] | 0.91 | 0.98 | 0.02 |

[a]SD, standard deviation of results based on a pooling of 3 days of feed.
[b]Distillers grain (corn) 30%, corn gluten meal 30%, canola meal 20%, and micronized soybean 20%,
[c]Crude fat: 80% min, free fatty acids: 1.5% max, fatty acids: 99.0% min, triglycerides: 3.0% max, unsaturated fatty acids: 46.0% max, moisture: 5.0% max, calcium: 8.5% min, and calcium salts of free fatty acids 80.0% min.
[d]Trace mineral supplement of control group contained per kilogram of dry matter: 12 mg of Co, 1,326 mg of Cu, 4,632 mg of Fe, 1,817 mg of Mn, and 6,428 mg of Zn, and trace mineral supplement of high trace mineral group contained per kilogram of dry matter: 64 mg of Co, 1,257 mg of Cu, 4,444 mg of Fe, 9,752 mg of Mn, and 14,454 mg of Zn.

In conclusion, the oversupplementation of minerals over a 3 week period had minimal impacts on the taxonomy of the gastrointestinal microbiome, indicating that levels of microbial toxicity were not reached and that microbial tolerance to dietary minerals is greater compared to that of pH tolerance. Differences between 16S rRNA amplicon sequencing microbial community profiles and cultured mixed isolates would indicate that there are biases depending on the method used and the method of sample collection. However, further work is needed to determine whether long-term exposure to heavy metals during lactation confers phenotypic AMR on the gastrointestinal microbiome. Even though no effect was discerned from the treatments provided in this study, there remain concerns about the accumulation of metals in soils as a result of multi-year manure amendments produced from cattle fed an oversupplementation of trace minerals.

## MATERIALS AND METHODS

### Experimental design

Cows were housed in a tie-stall facility at the experimental dairy farm of Agriculture and Agri-Food Canada (Sherbrooke, QC, Canada). Animals had been previously cannulated both in the rumen and duodenum. Selection was based on health status, age, stage of pregnancy, expected calving date, and previous number of lactations. The experiment was conducted with four rumen and closed T-shape duodenal (10–15 cm distal to the pylorus) cannulated multiparous dry non-pregnant Holstein dairy cows (779 ± 43 kg) in a 2 × 2 cross-over design ($n$ = 4 per treatment) over two periods of 26 days each. Animals were fed an isoenergetic, isonitrogenous lactation diet of about 60% forage and 40% concentrate with mineral supplementation based on body weight and production (650 kg cows, producing 40 kg of milk; Table 2; [4]) for control diet. Diet samples were collected on three consecutive days, pooled, and then subsampled for nutrient analysis. The high mineral diet included an increase in total mineral supplementation to reach a final concentration of zinc 0.89-fold higher compared to the CON based on common nutritional practices (5). Nutrient composition can be found in Table 3.

### Sample collection

The first 21 days of each period were for dietary adaptation and washout, and for the final 5 days, animals were sampled. Animals were fed 40 kg of total mixed ration once daily at 08:00 h, and water was provided *ad libitum*. Samples of digesta were collected from the rumen, duodenum, and rectum every 8 h during five consecutive days to

**TABLE 3** Total mixed ration nutrient composition for the basal diet for control (CON) and excess (HIGH) mineral supplementation[b]

| Item | CON | SD[a] | HIGH | SD |
|---|---|---|---|---|
| DM[a] (%) | 45.27 | 0.69 | 45.28 | 0.69 |
| CP (%) | 15.64 | 0.56 | 15.73 | 0.45 |
| ADF (%) | 19.40 | 0.83 | 19.50 | 0.83 |
| NDF (%) | 29.21 | 1.16 | 28.67 | 2.42 |
| Fat (%) | 3.83 | 0.00 | 3.86 | 0.00 |
| OM (%) | 93.56 | 0.41 | 93.47 | 0.35 |
| Starch (%) | 25.86 | 0.38 | 25.67 | 1.37 |
| Gross energy (Mcal/kg) | 4.44 | 0.00 | 4.44 | 0.00 |
| P (%) | 0.37 | 0.01 | 0.39 | 0.01 |
| K (%) | 1.30 | 0.04 | 1.30 | 0.04 |
| Ca (%) | 0.95 | 0.02 | 0.93 | 0.02 |
| Mg (%) | 0.28 | 0.01 | 0.27 | 0.01 |
| Fe (mg/kg) | 203.26 | 8.89 | 204.52 | 9.5 |
| Co (mg/kg) | 0.24 | 0.09 | 0.78 | 0.09 |
| Cu (mg/kg) | 19.21 | 4.60 | 18.73 | 1.63 |
| Mn (mg/kg) | 35.18 | 3.13 | 114.79 | 30.71 |
| Zn (mg/kg) | 89.14 | 4.40 | 168.27 | 28.65 |

[a]SD, standard deviation of analysis.
[b]DM, dry matter; CP, crude protein; ADF, acid detergent fiber; NDF, neutral detergent fiber; and OM, organic matter.

get a final data set representing each 2 h period of the day: 08:00, 10:00, 12:00, 14:00, 16:00, 18:00, 20:00, 22:00, 00:00, 02:00, 04:00, and 06:00 h. Samples for microbiology were taken at 0, 2, 4, 8, 12, 16, and 20 h post-feeding, where 08:00 is time zero, and feeding occurred immediately after the 00:00 h sampling. Rumen samples were taken by hand sampling at five points in the rumen via the cannula; medial, caudal-dorsal, caudal-ventral, cranial-ventral, and cranial-dorsal from liquid and solid digesta (35). The whole rumen content was then hand squeezed through four layers of cheesecloth to separate the liquid fraction. Duodenal content was collected via insertion of a bottle into the duodenal cannula to obtain 500 mL of content that was then filtered through four layers of cheesecloth. Fecal samples were collected by sterile handgrab. All samples were processed immediately. Samples from all niches were mixed and subsampled for immediate measurement of pH (Accumet pH Meter; Fisher Scientific, Waltham, MA, USA). For culturing analysis, 1 mL of digesta from each niche was added to 3.5 mL of 30% glycerol solution and then snap-frozen in liquid nitrogen to be stored at −80°C. For DNA extraction, 1.5 mL of content from each niche was snap-frozen in liquid nitrogen and stored at −80°C until processing.

## Determination of minimum inhibitory concentrations for zinc and copper

Specific strains, including *E. coli* ATCC 25922 and *Enterococcus faecalis* ATCC BAA-2128, were used as positive controls based on their known resistance to copper (36), as well as two *E. coli* (AN1-R2-P14-01 and AN1-R1-P01-01) isolates and a *Lelliottia amnigena* (AN-Abat-3L-A01) isolate from the in-house culture collection, which were used in the study based on the presence of copper and zinc resistance genes in their genomes. Following the procedure of Cidre et al. (37), we determined the MICs for copper and zinc using the CLSI (38) dilution antimicrobial susceptibility method with both microdilution plates and agar plates using Mueller-Hinton (BD Canada, Mississauga, ON, Canada) supplemented with $CuSO_4$ (Sigma Aldrich, MO, USA) in concentrations of 0, 8, 16 24, 32, and 48 mM, adjusted to pH 7.2, or $ZnCl_2$ (Sigma Aldrich, MO, USA) in concentrations of 0, 2, 4, 6, and 8 mM, adjusted to pH 7.2. Concentrations of metals in media were confirmed using the QuantiChrom copper and zinc assay kits (Bioassay System, USA). Briefly, the inoculated microdilution plates were incubated at 37°C for 18 h, and the absorbance was measured with spectrophotometry (SpectraMax ABS Plus, Molecular Devices, San

Jose, CA, USA). The same procedure was performed using agar plates, but incubation was done for intervals of 18 and 40 h (39). The first $CuSO_4$ or $ZnCl_2$ concentration without visible bacterial growth was considered the MIC for that metal (Table S1). The final results were 20 mM of $CuSO_4$ and 8 mM of $ZnCl_2$. All experiments were performed in duplicate.

## Culturing and phenotypic resistance tests

Duodenum, rumen, and fecal samples were plated on MacConkey agar in aerobic condition at 37°C for 24 h to isolate gram-negative enteric bacteria specifically *Enterobacteriaceae*. An extra step of pre-enrichment in 2% buffered peptone water at 37°C for 8 h in aerobic condition was performed with duodenum and rumen samples to balance pH and optimize bacterial growth prior to MacConkey plating. Mixed enriched isolates were made by diluting scraped colonies from three MacConkey plates in MacConkey broth with 30% glycerol and preserved at −80°C. MEIs were plated on Mueller-Hinton agar enriched with Cu (20 mM of $CuSO_4$), Zn (8 mM of $ZnCl_2$), and ceftriaxone (CRO) (4 mg/mL) separately and in all their combinations in aerobic conditions at 37°C for 24 h to evaluate the presence of resistance phenotype from cultivable gram-negative bacteria in the duodenum, rumen, and feces. The concentration used for the Cu, Zn, and CRO was determined by MIC using pooled fecal samples.

## DNA extraction and sequencing

DNA from ruminal and duodenal samples was extracted using 350 µL of sample and then mixed with 200 mL of 10% SDS and a release buffer containing 0.5% β-mercaptoethanol. They were processed using a double bead-beating method (3 min at 4,000 rpm) with an Omni Bead Ruptor 24 (PerkinElmer, Inc., Kennesaw, GA, USA) and separated by incubation for 15 min at 70°C. Salting out of the protein was achieved by adding 200 mL of 10 M sodium acetate solution. Purification of the DNA was done using the QIAamp Fast DNA Stool Mini Kit (Qiagen, Inc., Germantown, MD, USA) following the manufacturer's protocol.

DNA from feces was extracted using 0.5 g in 1 mL of 0.9% saline water followed by the same extraction and purification procedure as the rumen and duodenal liquid. All the eluates from the QIAamp Fast DNA Stool Mini Kit were additionally passed on One-Step PCR Inhibitor Removal Kit (Zymo Research, Inc., Irvine, CA, USA). DNA library preparation and sequencing were performed by an external laboratory (McGill University, Montreal, QC, Canada) as described by Franco-Lopez et al. (40). Briefly, the library was made by combining all the sample amplicons from a PCR using primers targeting the V4 region of the 16S rRNA gene (515F: GTGYCAGCMGCCGCGGTAA and 806R: GGACTACHVGGGTWTC-TAAT) with Illumina MiSeq paired-end sequencing platform. Sequences have been submitted to the NCBI sequence read archive under BioProject PRJNA1070637.

## Bioinformatics

Analysis for the sequence metadata was performed using R version 4.2.1 and the package DADA2 version 1.24.0. After quality inspection of raw paired sequences from the rumen ($n = 56$), duodenum ($n = 56$), and fecal ($n = 56$) samples, sequences were truncated at a length of 250 bp for the forward sequences and 200 bp for the reverse sequences to improve quality scores based on QIIME quality analysis following the QIIME2 SOP (41). Sequences were then filtered to remove paired sequences exceeding two errors per read. Then, an inference step was performed to remove substitutions, indel errors, and sequences that did not perfectly overlap with the target area. Amplicon sequence variants were taxonomically aligned using the SILVA version 138.1 database following the published procedure (42). Alpha diversity, Bray-Curtis distances, and phyla abundances were obtained using the Phyloseq package version 1.40.0. Beta diversity of the phyla was made using nonmetric multidimensional scaling, and a permutational analysis of variance was performed to obtain the *P* value between niches (rumen, duodenum, and feces). Sequences identifying members of the *Enterobacteriaceae*

family were specifically assessed for differences between treatments for comparison to culture-based analysis.

## Statistical analysis

The effects of the CON and HIGH dietary treatments on the niche's phyla, Shannon, and Simpson diversities were analyzed using a double-repeated measure over time ANOVA using unstructured and compound symmetry covariance structures for the niche and times of sampling. However, due to interactions, each niche was analyzed separately using the sp(pow) covariance structure for both indices. Comparisons of niches have been done with a Tukey adjustment on the means of dietary treatments within the niches for Shannon and Simpson diversities. Taxa abundance was not normally distributed; therefore, means were square transformed, and a PROC MIXED analysis was performed in SAS (9.4) for each niche. Culture results were analyzed for each niche through frequency tables over seven time points, which were considered replicates, with a Cochran-Mantel-Haenszel test. Statistical significance was declared with a $P < 0.05$, and a tendency at $0.05 < P < 0.10$.

## ACKNOWLEDGMENTS

We thank the government of Canada for funding support. The authors thank Steve Méthot from Agriculture and Agri-Food Canada, Sherbrooke Research and Development Centre for his help with the statistical analysis. Also, thanks to Geneviève Drouin for technical help with the sample collection and laboratory analysis. We thank all the barn staff for their help with the animal care and feeding trial and also the lab support of Dr. Ronholm's team from the Department of Animal Science, Faculty of Agricultural and Environmental Sciences, Macdonald Campus, McGill University for their help with the sequencing.

R.M.P. and M.D. designed and directed the project. R.M.P., M.D., and J.R. did the sequencing analysis. C.-A.M. processed the experimental data, performed the analysis, drafted the manuscript, and designed the figures. R.M.P. verified the methods. C.-A.M. and R.M.P. wrote the manuscript. M.D. and J.R. contributed to the final version of the manuscript.

## AUTHOR AFFILIATIONS

[1]Agriculture and Agri-Food Canada, Sherbrooke Research and Development Center, Sherbrooke, Quebec, Canada

[2]Animal Science Department, Faculty of Agriculture and Food Sciences, Laval University, Quebec City, Quebec, Canada

[3]Department of Food Science and Agricultural Chemistry, Faculty of Agricultural and Environmental Sciences, Macdonald Campus, McGill University, Montreal, Quebec, Canada

[4]Department of Animal Science, Faculty of Agricultural and Environmental Sciences, Macdonald Campus, McGill University, Montreal, Quebec, Canada

## AUTHOR ORCIDs

Jennifer Ronholm http://orcid.org/0000-0001-7902-3368
Renée M. Petri http://orcid.org/0000-0002-8209-8261

## ETHICS APPROVAL

Experimental procedures were approved by the Institutional Animal Care Committee of the Sherbrooke Research and Development Centre under the guidelines of the Canadian Council on Animal Care (2009).

## ADDITIONAL FILES

The following material is available online.

### Supplemental Material

**Table S1 (Spectrum01090-24-S0001.docx).** MIC concentration determination.

### Open Peer Review

**PEER REVIEW HISTORY (review-history.pdf).** An accounting of the reviewer comments and feedback.

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
