## [Reviewer comments · Microbiology Spectrum]

Microbiology Spectrum

The influence of trace metal supplementation on the presence of ceftriaxone resistance in Enterobacteriaceae in the gastrointestinal tract of dairy cattle

Charles-Antoine Martineau, Melissa Duplessis, Jennifer Ronholm, and Renee Petri

Corresponding Author(s): Renee Petri, Agriculture and Agri-Food Canada

Review Timeline:

Submission Date:	April 30, 2024
Editorial Decision:	July 29, 2024
Revision Received:	September 12, 2024
Editorial Decision:	October 18, 2024
Revision Received:	December 13, 2024
Accepted:	January 14, 2025

Editor: Bernadette Connors

Reviewer(s): Disclosure of reviewer identity is with reference to reviewer comments included in decision letter(s). The following individuals involved in review of your submission have agreed to reveal their identity: Alexandra Calle (Reviewer #1); Hassan M Al-Tameemi (Reviewer #2)

Transaction Report:

DOI: <https://doi.org/10.1128/spectrum.01090-24>

Re: Spectrum01090-24 (The influence of trace metal supplementation on the presence of ceftriaxone resistance in Enterobacteriaceae in the gastrointestinal tract of dairy cattle)

Dear Dr. Renee Maxine Petri:

Thank you for the privilege of reviewing your work. Below you will find my comments, instructions from the Spectrum editorial office, and the reviewer comments.

Revision Guidelines

Sincerely,
Bernadette Connors
Editor
Microbiology Spectrum

Reviewer #1 (Comments for the Author):

The paper is well written. However, I think the analysis of the results and even the methods are too simple. I question the use of 4 animals for the treatment; however, I understand the complexity of expanding. These are my comments and recommendations:

- The use of only 4 animals per treatment seems to be a small sample for drawing conclusions. It would be beneficial to explain

how you calculated the sample size and why you believe that 4 animals per treatment are sufficient for drawing conclusions.

- Could you provide more details about the criteria used for selecting the animals? A more comprehensive explanation in the experimental design section would be helpful.

- Line 284. A period is missing after (2009)

- Line 315 Culturing methods, Figure 6, and results related to that portion of the experiment: What was the purpose of testing any gram-negative bacteria? This is a very wide group of organisms. You should inform your readers why you targeted them and for what purpose. For the results, I recommend that you explain which microorganisms you think you are enumerating. I know you chose gram neg, but it seems too broad to me. The results presented for this portion are way too simple, as there is no information that can help with any significant conclusion. For the graphs, you also need to report standard deviations and error bars. Are the results average for all samples? How different were the samples between them? For this same part of your research, I noticed in the discussion that you described conducting additional analysis.

Lines 250-253 indicate you plated on selective media. Please add this to the methods, describe the media used, and the conditions of this experiment. Present the results of this experiment with statistical analysis. Line 252, you say "no significant effect of the diet..." please provide statistics apply so you can use the word "significant"

- Lastly, please correlate your findings in the microbiota analy with culturing results.

Reviewer #2 (Comments for the Author):

This study examined the microbiota in the rumen, duodenum, and feces to investigate the development and dissemination of resistance to copper, zinc, and ceftriaxone in the digestive system when there is an excessive supply of trace elements in the diet. The researchers hypothesized that excessive zinc supplementation in the diet would modify the microbial population in various segments of the gastrointestinal system, resulting in an increased resistance to copper, zinc, and ceftriaxone. The researchers concluded that although there was a reduction in microbial diversity in the HIGH treatment, the number of enterobacteriaceae remained the same. Excessive zinc supplementation had a negligible impact on the ability of the gut microbiota to resist. The work is well written, and the experiments are well designed. This study addressed an important component of antimicrobial resistance (AMR) and its relationship with transition metals.

Remarks:

1. The authors argued that the lack of effect of zinc and copper on the AMR profile could be due to the fact that metal concentrations in the gut had not yet reached a toxic level for bacteria, and that PH could potentially reduce the toxicity of metals or their toxic forms. Despite these points' validity, the study failed to measure and characterize the metal concentrations in the rumen or duodenum samples. This would have validated the explanations provided by the authors. Furthermore, plating the samples and numerating bacteria could be misleading, as they may favor the growth of some bacteria over others.

2. The experiment's timeframe for resistance evolution was short. Studies that demonstrated strong correlation between metal exposition and resistance come from environmental sampling where the microorganisms are in contact with high level of metals for extended period of time. The concurrent existence of metal resistance genes (MRGs) and antibiotic resistance genes (ARGs) is a result of an exchange of genetic components. MRGs and ARGs are frequently found together on the same plasmids, transposons, or integrons. This co-occurrence is facilitated by cross-resistance mechanisms and is an indication of their interconnected evolutionary history. Metals and antibiotics can exhibit cross-resistance mechanisms, enabling bacteria to adapt through similar physiological mechanisms. The presence of shared resistance mechanisms implies that these genes have undergone co-evolutionary processes across time. Hence, prolonging the period of the experiment could have yielded a contrasting result. Moreover, I am wondering about the necessity of combining the metals with feed and holding the mixture for an extended duration to enhance co-selection mechanisms and impact the outcomes of this study. This would ensure a higher and more prolonged exposure compared to the digestive system.

3. Line 9 tables: "Trace mineral supplement of control group contained per kilogram of dry matter: 12 mg of Co, 1,326 mg of Cu, 4,632 mg of Fe, 1,817 mg of Mn, and 6,428 mg of Zn, and trace mineral supplement of high trace mineral group contained per kilogram of dry matter: 12 mg of Co, 1,257 mg of Cu, 4,444 mg of Fe, 9,752 mg of Mn, and 14,454 mg of Zn. Why was the concentration of copper in the control group (1,326) higher than that in the high trace mineral group (1,257)? Why have manganese and cobalt increased in the high-trace mineral group, given that zinc and copper are the focus of this study? .

Brodie F Gillieatt, Nicholas V Coleman, Unravelling the mechanisms of antibiotic and heavy metal resistance co-selection in environmental bacteria, *FEMS Microbiology Reviews*, 2024;, fuae017, <https://doi.org/10.1093/femsre/fuae017>

Chen J, Li J, Zhang H, Shi W, Liu Y. Bacterial Heavy-Metal and Antibiotic Resistance Genes in a Copper Tailing Dam Area in Northern China. *Front Microbiol*. 2019 Aug 20;10:1916. doi: 10.3389/fmicb.2019.01916. PMID: 31481945; PMCID: PMC6710345.

Singh, C.K., Sodhi, K.K., Shree, P. et al. Heavy Metals as Catalysts in the Evolution of Antimicrobial Resistance and the Mechanisms Underpinning Co-selection. *Curr Microbiol* 81, 148 (2024). <https://doi.org/10.1007/s00284-024-03648-2>

Peng S, Zheng H, Herrero-Fresno A, Olsen JE, Dalsgaard A, Ding Z. Co-occurrence of antimicrobial and metal resistance genes in pig feces and agricultural fields fertilized with slurry. *Sci Total Environ*. 2021 Oct 20;792:148259. doi: 10.1016/j.scitotenv.2021.148259. Epub 2021 Jun 17. PMID: 34147788.

Reviewer #3 (Comments for the Author):

Comments for authors

General comments

The authors conducted a research to assess the influence of trace metal supplementation on the presence of ceftriaxone resistance in Enterobacteriaceae in the gastrointestinal tract of dairy cattle. This is an interest research area due to the impact of antibiotic use in livestock farming and AMR dissemination in the environment. The manuscript needs to be proofread for typographical and grammatical errors. There are some bits of the study design that need further details and explanation. Some conflicting values need clarification.

Specific comments

1. Abstract: In line 25-31, it is important to be consistent about what metals were targeted for the supplementation and assessed. In some parts of the manuscript, only Zinc was mentioned and in other parts, Zinc and copper were mentioned. Similarly, In Line 291, the authors have mentioned "The concentrations of minerals formulated in the high mineral diet (HIGH) were 3.25, 3.26, and 1.88 fold higher for Co, Mn, and Zn, respectively, compared to the CON"
2. In the abstract, it states in line 27 "surplus levels of dietary zinc (2.5x; HIGH) compared to the recommended levels (CON).. This 2.5x mentioned is different from the 1.88 fold above for Zinc.
3. It is important that measurement of the ingredient composition of the diets is carried out in replicate. If this was done, are the values presented in Tables 1 and 2 the mean values? If yes, please add the standard deviations. If not, this could be another limitation of the study.
4. Was the microbiome of the cattle assessed before initiating the diet in the control and treatment groups to know the relative abundance of different resident bacterial groups in the cattle? This should be clearly stated if this was done. If not. This could be an important factor that can affect the microbial diversity.
5. Also was the heavy metal tolerance and ceftriaxone resistance pattern of resident bacteria known before conducting the experiment? It is also possible the heavy metal and antibiotic coresistance may have existed in resident bacteria before the experiment.
6. Figure 1 is titled "Daily variation in spot-sampled pH for each niche (rumen, duodenum, and faeces) relative to the number of hour post feeding". However, the pH values presented are for 20 hours.
7. Table 3 has an asterisk in the final column without a footnote to explain this.
8. Line 322-326 states: MEI were plated on Mueller Hinton agar enriched with Cu (20 mM of CuSO₄), Zn (8 mM of ZnCl₂), and Ceftriaxone (CRO) (4 mg/ml) separately and all their combinations in aerobic condition at 37 C for 24 hours to evaluate the presence of resistance phenotype from cultivable Gram-negative bacteria in the duodenum, rumen and faeces. The results in Figure 6 has only presented each individual selective media for copper (Cu; blue), zinc (Zn; red), and ceftriaxone (CRO; green) without their combinations as mentioned above.
9. Line 292-294: states that "The concentrations of minerals formulated in the high mineral diet (HIGH) were 3.25, 3.26, and 1.88 fold higher for Co, Mn, and Zn, respectively, compared to the CON (Tables 1 and 2).
 - Please include the value of the folds for Copper since it was one of the metals of interest.
 - Also, the footnote on Table 2 provides this details.....Trace mineral supplement of control group contained per kilogram of dry matter: 12 mg of Co, 1,326 mg of Cu, 4,632 mg of Fe, 1,817 mg of Mn, and 6,428 mg of Zn, and trace mineral supplement of high trace mineral group contained per kilogram of dry matter: 64 mg of Co, 1,257 mg of Cu, 4,444 mg of Fe, 9,752 mg of Mn, and 14,454 mg of Zn.Based on this information, it is not clear how the number of folds of the minerals was calculated in Line 292-294.

Comments for authors

General comments

The authors conducted a research to assess the influence of trace metal supplementation on the presence of ceftriaxone resistance in Enterobacteriaceae in the gastrointestinal tract of dairy cattle. This is an interest research area due to the impact of antibiotic use in livestock farming and AMR dissemination in the environment. The manuscript needs to be proofread for typographical and grammatical errors. There are some bits of the study design that need further details and explanation. Some conflicting values need clarification.

Specific comments

1. Abstract: In line 25-31, it is important to be consistent about what metals were targeted for the supplementation and assessed. In some parts of the manuscript, only Zinc was mentioned and in other parts, Zinc and copper were mentioned. Similarly, In Line 291, the authors have mentioned “The concentrations of minerals formulated in the high mineral diet (HIGH) were 3.25, 3.26, and **1.88 fold** higher for Co, Mn, and Zn, respectively, compared to the CON”
2. In the abstract, it states in line 27 “surplus levels of dietary zinc (**2.5x**; HIGH) compared to the recommended levels (CON).. This 2.5x mentioned is different from the 1.88 fold above for Zinc.
3. It is important that measurement of the ingredient composition of the diets is carried out in replicate. If this was done, are the values presented in Tables 1 and 2 the mean values? If yes, please add the standard deviations. If not, this could be another limitation of the study.
4. Was the microbiome of the cattle assessed before initiating the diet in the control and treatment groups to know the relative abundance of different resident bacterial groups in the cattle? This should be clearly stated if this was done. If not. This could be an important factor that can affect the microbial diversity.
5. Also was the heavy metal tolerance and ceftriaxone resistance pattern of resident bacteria known before conducting the experiment? It is also possible the heavy metal and antibiotic coresistance may have existed in resident bacteria before the experiment.
6. Figure 1 is titled “Daily variation in spot-sampled pH for each niche (rumen, duodenum, and faeces) relative to the number of hour post feeding”. However, the pH values presented are for 20 hours.
7. Table 3 has an asterisk in the final column without a footnote to explain this.
8. Line 322-326 states: MEI were plated on Mueller Hinton agar enriched with Cu (20 mM of CuSO₄), Zn (8 mM of ZnCl₂), and Ceftriaxone (CRO) (4 mg/ml) **separately and all their combinations** in aerobic condition at 37°C for 24 hours to evaluate the presence of resistance phenotype from cultivable Gram-negative bacteria in the duodenum, rumen and faeces.

The results in Figure 6 has only presented each individual selective media for copper (Cu; blue), zinc (Zn; red), and ceftriaxone (CRO; green) without their combinations as mentioned above.

9. Line 292-294: states that “The concentrations of minerals formulated in the high mineral diet (HIGH) were 3.25, 3.26, and 1.88 fold **higher for Co, Mn, and Zn**, respectively, compared to the CON (Tables 1 and 2).

- Please include the value of the folds for Copper since it was one of the metals of interest.
- Also, the footnote on Table 2 provides this details.....Trace mineral supplement of control group contained per kilogram of dry matter: 12 mg of Co, 1,326 mg of Cu, 4,632 mg of Fe, 1,817 mg of Mn, and 6,428 mg of Zn, and trace mineral supplement of high trace mineral group contained per kilogram of dry matter: 64 mg of Co, 1,257 mg of Cu, 4,444 mg of Fe, 9,752 mg of Mn, and 14,454 mg of Zn.

Based on this information, it is not clear how the number of folds of the minerals was calculated in Line 292-294.

Response to reviewers

Reviewer #1 (Comments for the Author):

The paper is well written. However, I think the analysis of the results and even the methods are too simple. I question the use of 4 animals for the treatment; however, I understand the complexity of expanding. These are my comments and recommendations:

Response: Thank you for your comment, the authors recognize the limitations of using only 4 animals (see comments below) and can appreciate that there are more complex analyses that could be performed. However, in this experiment, the authors choose to take repeated samples on 4 animals to increase accuracy and repeatability which means fewer animals and lower standard deviation due to inter-animal variation in the microbiomes (Petri et al., 2013a). The goal of this paper is to determine if there is a causality between short-term increases in dietary minerals and changes in the microbiome of the fore, mid and hindgut populations of the dairy cow and if these changes correspond to variation in microbial tolerance to heavy metals. This was undertaken with the idea that if differences were seen, then a more targeted (single niche) and more in-depth (metagenomic) analysis could be performed on a larger cohort of animals.

- The use of only 4 animals per treatment seems to be a small sample for drawing conclusions. It would be beneficial to explain how you calculated the sample size and why you believe that 4 animals per treatment are sufficient for drawing conclusions.

Response: The authors understand that the use of a statistically incorrect sample size may lead to inadequate results. In the current study, a power analysis was performed using pH values for the duodenum as those were the only values available from previous research on these same animals. The effective sample size was $n=2.8$ per group. Animals were already prechosen and pre-fitted for the duodenal cannulas, therefore to account for low animal numbers, the study including several time replicates as well as multiple gastrointestinal niches as repeated measures to increase statistical power. While repeated measures does not increase the accuracy of the data due to inter-animal variation, it does increase the sample size.

Power of alpha diversity was run post-hoc to provide context for the reviewers – power values were .92 for Shannon. However, the authors would like to acknowledge this issue with variation in effective sample size based on different types of data in the experimental design and have included the following in lines 197 - 199:

- Could you provide more details about the criteria used for selecting the animals? A more comprehensive explanation in the experimental design section would be helpful.

Response: Animals were previously selected for duodenal cannulation surgery based on good health records, similar ages, similar calving dates, and lactation number. These animals were already available and the reuse of animals with duodenal cannulas was important as these are difficult surgeries to complete and care for.

This information has been added in lines 294-296

- Line 284. A period is missing after (2009)

Thank you for seeing this, the period has been added.

- Line 315 Culturing methods, Figure 6, and results related to that portion of the experiment: What was the purpose of testing any gram-negative bacteria? This is a very wide group of organisms. You should inform your readers why you targeted them and for what purpose.

Response: Thank you for this comment, the authors agree and have added this information in at lines 71-76.

“Enterobacteriaceae are Gram-negative bacteria that are commonly used as viable indicators for fecal contamination and are identified as priority pathogens for public health (WHO, 2024). They are also known to be hosts of clinically important antibiotic resistance genes, as well as being identified as having metal resistance genes (Cufaoglu et al., 2022) making them excellent candidates for understanding the transmission of AMR.”

For the results, I recommend that you explain which microorganisms you think you are enumerating. I know you chose gram neg, but it seems too broad to me. The results presented for this portion are way too simple, as there is no information that can help with any significant conclusion.

Response: Thank you for this comment, the idea here is to see if we find resistance that could be associated with *Enterobacteriaceae*. Specifically, *E. coli*, *Enterobacteria*, *Klebsiella*, *Salmonella*, *Proteus*, *Yersinia*, and *Pseudomonas*. All of which are considered bacteria of importance for human health. This is why we focused on the order *Enterobacteriales* in the 16S data to potentially indicate which groups could be associated with those grown on MacConkey. This information has been added to lines 255-258 of the discussion and lines 329-330 of the materials and methods.

For the graphs, you also need to report standard deviations and error bars. Are the results average for all samples? How different were the samples between them? For this same part of your research, I noticed in the discussion that you described conducting additional analysis.

Response: Thank you for this comment. For relative abundance and alpha diversity plots, there is not generally standard deviation published in the graphics, so the authors have decided to not try to adjust the graphics to add them. However, for Figure 1 there was already error bars, and for Figure 6 the authors have added these.

Lines 250-253 indicate you plated on selective media. Please add this to the methods, describe the media used, and the conditions of this experiment. Present the results of this experiment with statistical analysis. Line 252, you say "no significant effect of the diet..." please provide statistics apply so you can use the word "significant"

Response: Thank you for your comments. This information is provided in the Culturing section of the materials and methods, the authors discuss selective media with reference to the plating of mixed enriched isolates with copper, zinc and ceftriaxone (Lines 330-340). While we appreciate the reviewers statement about statistics and the word significant, these results have been stated in the results section under “Culturing” (Lines 164-168) and the authors prefer not to repeat p-values in the discussion.

- Lastly, please correlate your findings in the microbiota analysis with culturing results.

Response: Thank you for this comment, the authors agree this would be a fabulous combination, but due to the necessity to selectively culture, the plating results are not equivalent to the whole microbiome data presented with 16S. Since the 2 methods have their own biases, the authors do not wish to over interpret the results through correlation. However, we have added a statement to this factor in Lines 273-276.

Reviewer #2 (Comments for the Author):

This study examined the microbiota in the rumen, duodenum, and feces to investigate the development and dissemination of resistance to copper, zinc, and ceftriaxone in the digestive system when there is an excessive supply of trace elements in the diet. The researchers hypothesized that excessive zinc supplementation in the diet would modify the microbial population in various segments of the gastrointestinal system, resulting in an increased resistance to copper, zinc, and ceftriaxone. The researchers concluded that although there was a reduction in microbial diversity in the HIGH treatment, the number of enterobacteriaceae remained the same. Excessive zinc supplementation had a negligible impact on the ability of the gut microbiota to resist. The work is well written, and the experiments are well designed. This study addressed an important component of antimicrobial resistance (AMR) and its relationship with transition metals.

Remarks:

1. The authors argued that the lack of effect of zinc and copper on the AMR profile could be due to the fact that metal concentrations in the gut had not yet reached a toxic level for bacteria, and that PH could potentially reduce the toxicity of metals or their toxic forms. Despite these points' validity, the study failed to measure and characterize the metal concentrations in the rumen or duodenum samples. This would have validated the explanations provided by the authors. Furthermore, plating the samples and numerating bacteria could be misleading, as they may favor the growth of some bacteria over others.

Response: Thank you for this comment, the authors absolutely agree with these statements. Due to cost limitations, it was only possible to measure mineral levels in the feces and feed, hence why this is the authors speculation as to the potential reason for these differences. The authors recognize that plating and numerating a selected group of bacteria could be misleading, the authors have focused on specifically only Enterococcaceae due to the human health component and therefore the growth of only a select few well characterized bacteria to limit over interpretation and errors in bacterial counts.

2. The experiment's timeframe for resistance evolution was short. Studies that demonstrated strong correlation between metal exposition and resistance come from environmental sampling where the microorganisms are in contact with high level of metals for extended period of time.

The concurrent existence of metal resistance genes (MRGs) and antibiotic resistance genes (ARGs) is a result of an exchange of genetic components. MRGs and ARGs are frequently found together on the same plasmids, transposons, or integrons. This co-occurrence is facilitated by cross-resistance mechanisms and is an indication of their interconnected evolutionary history. Metals and antibiotics can exhibit cross-resistance mechanisms, enabling bacteria to adapt through similar physiological mechanisms. The presence of shared resistance mechanisms implies that these genes have undergone co-evolutionary processes across time. Hence, prolonging the period of the experiment could have yielded a contrasting result. Moreover, I am wondering about the necessity of combining the metals with feed and holding the mixture for an extended duration to enhance co-selection mechanisms and impact the outcomes of this study. This would ensure a higher and more prolonged exposure compared to the digestive system.

Response: Thank you for this comment. The authors agree that a more extended period would be helpful. However, due to changes in animal diets throughout their reproductive/production cycles, the study was looking to determine if there is “already” a short term indication of bacterial shifts that could indicate the increased presence of metal resistance genes (due to environmental adaptation) or of resistant groups. Since heavy metals such as zinc are a common component of the animal diets, this study was designed to look if short-term excessive exposure is able to initiate these changes. As metal resistance genes have been found to be co-localized with resistance genes in other species of livestock potentially due to management practices (Poulin-Laprade et al. 2021). To support this, the authors have added this reference to line 67-68.

3. Line 9 tables: "Trace mineral supplement of control group contained per kilogram of dry matter: 12 mg of Co, 1,326 mg of Cu, 4,632 mg of Fe, 1,817 mg of Mn, and 6,428 mg of Zn, and trace mineral supplement of high trace mineral group contained per kilogram of dry matter: 12 mg of Co, 1,257 mg of Cu, 4,444 mg of Fe, 9,752 mg of Mn, and 14,454 mg of Zn. Why was the concentration of copper in the control group (1,326) higher than that in the high trace mineral group (1,257)? Why have manganese and cobalt increased in the high-trace mineral group, given that zinc and copper are the focus of this study? .

Response: Thank you for this comment, the authors provided the data from NIMR analysis of the diets done after the experiment. Due to the levels of detection, the standard deviation in copper levels was too high to detect differences. Due to the nature of the mineral pre-mix, the levels of manganese and cobalt increased as well, but were not studied in this experiment as they are not known for increasing MRG presence in bacteria at this time.

Beaber JW, Hochhut B, Waldor MK. SOS response promotes horizontal dissemination of antibiotic resistance genes. *Nature*. 2004 Jan 1;427(6969):72-4. doi: 10.1038/nature02241. Epub 2003 Dec 21. PMID: 14688795.

Brodie F Gillieatt, Nicholas V Coleman, Unravelling the mechanisms of antibiotic and heavy metal resistance co-selection in environmental bacteria, FEMS Microbiology Reviews, 2024; fuae017, <https://doi.org/10.1093/femsre/fuae017>

Chen J, Li J, Zhang H, Shi W, Liu Y. Bacterial Heavy-Metal and Antibiotic Resistance Genes in a Copper Tailing Dam Area in Northern China. Front Microbiol. 2019 Aug 20;10:1916. doi: 10.3389/fmicb.2019.01916. PMID: 31481945; PMCID: PMC6710345.

Singh, C.K., Sodhi, K.K., Shree, P. et al. Heavy Metals as Catalysts in the Evolution of Antimicrobial Resistance and the Mechanisms Underpinning Co-selection. Curr Microbiol 81, 148 (2024). <https://doi.org/10.1007/s00284-024-03648-2>

Peng S, Zheng H, Herrero-Fresno A, Olsen JE, Dalsgaard A, Ding Z. Co-occurrence of antimicrobial and metal resistance genes in pig feces and agricultural fields fertilized with slurry. Sci Total Environ. 2021 Oct 20;792:148259. doi: 10.1016/j.scitotenv.2021.148259. Epub 2021 Jun 17. PMID: 34147788.

The authors thank the reviewer for the addition of these references for our review. After examining these, we have agreed that they are matching to those references we currently have and the list of current references is already substantial. As such we have chosen not to add these references to the current list in the manuscript.

Reviewer 3

Comments for authors General comments The authors conducted a research to assess the influence of trace metal supplementation on the presence of ceftriaxone resistance in Enterobacteriaceae in the gastrointestinal tract of dairy cattle. This is an interest research area due to the impact of antibiotic use in livestock farming and AMR dissemination in the environment. The manuscript needs to be proofread for typographical and grammatical errors. There are some bits of the study design that need further details and explanation. Some conflicting values need clarification.

Specific comments

1. Abstract: In line 25-31, it is important to be consistent about what metals were targeted for the supplementation and assessed. In some parts of the manuscript, only Zinc was mentioned and in other parts, Zinc and copper were mentioned. Similarly, In Line 291, the authors have mentioned "The concentrations of minerals formulated in the high mineral diet (HIGH) were 3.25, 3.26, and 1.88 fold higher for Co, Mn, and Zn, respectively, compared to the CON"

Response: Thank you for your assistance in improving this manuscript. The authors understand the confusion and have attempted to address this in a number of ways. Firstly, we have altered line 99 to be more clear that heavy metal increase in the diet was zinc, but that culturing was

both zinc and copper. As the experiment was designed to match standard animal husbandry practices, the authors used a commercial mineral supplement at 2 different feeding levels. Therefore the total mineral concentration was higher; however, there is no literature to support the impact of Co, and Mn as impacting resistance and authors focused only on zinc in the diet. The authors also assessed copper resistance in the selective media as copper is a known metal for resistance development and a common component of the dairy industry (foot baths and teat dips as well as dietary). Unfortunately, this could not be assessed as part of the diet since the levels were not higher in the HIGH compared to the control. We have added/edited information in lines 293-297 to try to address this.

2. In the abstract, it states in line 27 “surplus levels of dietary zinc (2.5x; HIGH) compared to the recommended levels (CON).. This 2.5x mentioned is different from the 1.88 fold above for Zinc.

Response: Thank you very much for finding these errors, it has been corrected to 0.89 based on the final dietary calculations done and shown in Table 2. .

3. It is important that measurement of the ingredient composition of the diets is carried out in replicate. If this was done, are the values presented in Tables 1 and 2 the mean values? If yes, please add the standard deviations. If not, this could be another limitation of the study.

Response: Thank you for your comment, the diets were collected 3 days in a row during the sampling collection. SD were the same between treatments for Table 1. Then, rations from the 3 days were pooled together and analyzed in duplicate. As usual, CV < 5 or 10 % depending on the analysis were accepted. Hence, we obtained nutrient composition on only 4 diets (HIGH, CON of period 1 and HIGH and CON of period 2). This information has been added to line 303-304 and the SD have been added to both tables.

Was the microbiome of the cattle assessed before initiating the diet in the control and treatment groups to know the relative abundance of different resident bacterial groups in the cattle? This should be clearly stated if this was done. If not. This could be an important factor that can affect the microbial diversity.

Response: Thank you for this comment, unfortunately, microbiome was not assessed prior to the experiment. These cattle were chosen based on their unique cannulation and good health status. Since their cannulation could potentially impact their baseline microbiomes, animals were used in a latin square design with a 3 week washout period to ensure that “baseline” gut microbiomes could be reestablished between periods. While baseline microbiomes could be an impacting factor for microbial diversity, analysis of the bacterial populations in this study do not show deviation from generally published data.

4. Also was the heavy metal tolerance and ceftriaxone resistance pattern of resident bacteria known before conducting the experiment? It is also possible the heavy metal and antibiotic coresistance may have existed in resident bacteria before the experiment.

Response: Thank you for your comments. Animals were not previously exposed to ceftriazone and therefore, resistance was assessed to understand if there is a co-development. As animals are fed heavy metals as part of a standard diet, the “baseline” potential for resistance would be equivalent to our “control” group and therefore, was not assessed prior to the experiment. We

agree that co-resistance could have previously existed, but this would be equivalent to what was seen in our control group.

6. Figure 1 is titled “Daily variation in spot-sampled pH for each niche (rumen, duodenum, and faeces) relative to the number of hour post feeding”. However, the pH values presented are for 20 hours.

Response: Thank you for your comment. As the sampling was done at each of the times indicated by points in the graphic, the 24h and the 0h are essentially the same sample. As such the graphic ends with the final pH taken at 20h post feeding.

7. Table 3 has an asterisk in the final column without a footnote to explain this.

Response: Thank you for catching this, this asterisk was supposed to indicate the interaction – an appropriate symbol as been used to replace the asterisk.

8. Line 322-326 states: MEI were plated on Mueller Hinton agar enriched with Cu (20 mM of CuSO₄), Zn (8 mM of ZnCl₂), and Ceftriaxone (CRO) (4 mg/ml) separately and all their combinations in aerobic condition at 37°C for 24 hours to evaluate the presence of resistance phenotype from cultivable Gram-negative bacteria in the duodenum, rumen and faeces. The results in Figure 6 has only presented each individual selective media for copper (Cu; blue), zinc (Zn; red), and ceftriaxone (CRO; green) without their combinations as mentioned above.

Response: Thank you for making note of our missing statement. There was no growth on the combination plates and therefore, they were not included in figure 6. We have addressed this by adding this statement to 164-165.

9. Line 292-294: states that “The concentrations of minerals formulated in the high mineral diet (HIGH) were 3.25, 3.26, and 1.88 fold higher for Co, Mn, and Zn, respectively, compared to the CON (Tables 1 and 2). □ Please include the value of the folds for Copper since it was one of the metals of interest.

Response: Thank you for your comment. As this sentence has been corrected to more accurately reflect the change in zinc, as the primary focus of the study, we have not added the copper values here. However, these values remain in Table 2 for reference.

□ Also, the footnote on Table 2 provides this details.....Trace mineral supplement of control group contained per kilogram of dry matter: 12 mg of Co, 1,326 mg of Cu, 4,632 mg of Fe, 1,817 mg of Mn, and 6,428 mg of Zn, and trace mineral supplement of high trace mineral group contained per kilogram of dry matter: 64 mg of Co, 1,257 mg of Cu, 4,444 mg of Fe, 9,752 mg of Mn, and 14,454 mg of Zn.

Based on this information, it is not clear how the number of folds of the minerals was calculated in Line 292-294.

Response: Thank you for your comment, the calculation was performed using the analyzed zinc content in the final rations that were shown in table 2.

Re: Spectrum01090-24R1 (The influence of trace metal supplementation on the presence of ceftriaxone resistance in Enterobacteriaceae in the gastrointestinal tract of dairy cattle)

Dear Dr. Renee Maxine Petri:

Thank you for the privilege of reviewing your work. Below you will find my comments, instructions from the Spectrum editorial office, and the reviewer comments.

Revision Guidelines

Sincerely,
Bernadette Connors
Editor
Microbiology Spectrum

Reviewer #1 (Comments for the Author):

Line 57: 20 000 kg. Replace with 20,000 kg.

Line 86-87: you indicated that resistance to ceftriaxone and other third-generation cephalosporins is increasing in Salmonella. You need a citation here to support your argument. Next, on line 88 to indicate that susceptibility in E. coli has been reported.

Thus, since you were talking about salmonella, please make that paragraph more cohesive, and make sure you add sources to support the statement about Salmonella resistant to ceftriaxone.

Line 109: delete the extra period at the end of the sentence.

Lines 310-312 AND connected to lines 104-105 (objectives): "The high mineral diet (HIGH) included an increase in total mineral supplementation to reach a final concentration of zinc in the 0.89 fold higher compared to the CON based on common nutritional practices (Tables 1 and 2; 5)."

Did you enrich with the other two (Cu and CRO)? This is confusing because your objectives indicated that you wanted to assess the potential emergence of resistance to copper, zinc, and ceftriaxone (104-105). Then, in methods (310-312) you only mention enriching with zinc. Further in methods, you indicated that you enriched McConkey media with Cu, Zn, and CRO. How did you measure the effect of the three elements?? Even though tables 1 and 2 mention the composition of the diets, that part is not clearly explained in methods.

Line 312: make sure you mention what information is expected in Tables 1 and 2

Lines 344-346: "The concentration used for the Cu, Zn, and CRO were determined by minimal inhibition concentration (MIC) using pooled faeces samples."

If you estimate the MIC, please add this to the methods portion. If you obtained MIC from a source, please reference that source." In other words, specify how you came up with the concentrations used for Cu, Zn, and CRO.

Results: because your objectives state that you wanted to assess the potential emergence of resistance to copper, zinc, and ceftriaxone (104-105), it is expected that the results are consistent with that. I did not find between lines 112 to 163 any evidence of accomplishing the objective. Culturing mentions the word resistance, but culturing is not a means to assess "antimicrobial resistance." It would be best if you clarified how you met the objectives.

Objectives (104-106): how did you "determine the potential emergence of resistance to copper, zinc, and ceftriaxone based on dietary mineral supplementation commonly found in commercial dairy farms.? Which specific portion of your methods measured the emergency of resistant bacteria? Please justify that whatever method was used is a valid method for that purpose.

Reviewer #2 (Comments for the Author):

Thank you for your reply to my comments. In your discussion or introduction, please clearly highlight the effect of extended periods of metals exposure vs. short-term exposure on resistance genes. Animal feed processors may misinterpret your study's conclusions, so it's crucial to emphasize this difference.

Please note that authors have addressed my comments, but I still feel that there is a need to reword the findings. I propose rephrasing the conclusions as follows: "At the conditions of our experiments, short exposure to minerals does not affect AMR." If the authors established prolonged exposure to metals, then their statement is incorrect. This is in contrast to the current conclusion: "In conclusion, the oversupplementation of minerals had minimal impacts on the gastrointestinal microbiome, indicating that levels of microbial toxicity were not reached..."

Response to reviewers

Reviewer #1 (Comments for the Author):

Line 57: 20 000 kg. Replace with 20,000 kg.

Thank you very much. This has been completed.

Line 86-87: you indicated that resistance to ceftriaxone and other third-generation cephalosporins is increasing in Salmonella. You need a citation here to support your argument. Next, on line 88 to indicate that susceptibility in E. coli has been reported. Thus, since you were talking about salmonella, please make that paragraph more cohesive, and make sure you add sources to support the statement about Salmonella resistant to ceftriaxone.

Thank you for your comment. In the reference provided in line 83, it states that resistance is increasing in Salmonella is increasing. Since we have already cited this paper in the sentence prior to the one you are asking a reference for, we do not wish to repeat the reference, but instead have adjusted the text for clarity. Please see lines 81-90.

Line 109: delete the extra period at the end of the sentence.

Thank you very much. This has been completed.

Lines 310-312 AND connected to lines 104-105 (objectives): "The high mineral diet (HIGH) included an increase in total mineral supplementation to reach a final concentration of zinc in the 0.89 fold higher compared to the CON based on common nutritional practices (Tables 1 and 2; 5)."

Did you enrich with the other two (Cu and CRO)? This is confusing because your objectives indicated that you wanted to assess the potential emergence of resistance to copper, zinc, and ceftriaxone (104-105). Then, in methods (310-312) you only mention enriching with zinc. Further in methods, you indicated that you enriched McConkey media with Cu, Zn, and CRO. How did you measure the effect of the three elements?? Even though tables 1 and 2 mention the composition of the diets, that part is not clearly explained in methods.

Thank you for your comment. We did have the objective to assess the impact of increased zinc on general heavy metal resistance, and antibiotic resistance based on co-localization of the MRG and ARG found in other studies as stated in the introduction. We did not enrich for Cu or CRO. Therefore we have adjusted our objectives in lines 102-104 to try to capture this. ,

Line 312: make sure you mention what information is expected in Tables 1 and 2

Thank you, this has been adjusted in lines 303-308

Lines 344-346: "The concentration used for the Cu, Zn, and CRO were determined by minimal inhibition concentration (MIC) using pooled faeces samples."

If you estimate the MIC, please add this to the methods portion. If you obtained MIC from

a source, please reference that source." In other words, specify how you came up with the concentrations used for Cu, Zn, and CRO.

Thank you for this comment. We have added the determination methods and a supplementary table with results. Please see lines 333-351.

Results: because your objectives state that you wanted to assess the potential emergence of resistance to copper, zinc, and ceftriaxone (104-105), it is expected that the results are consistent with that. I did not find between lines 112 to 163 any evidence of accomplishing the objective. Culturing mentions the word resistance, but culturing is not a means to assess "antimicrobial resistance." It would be best if you clarified how you met the objectives.

Thank you, please see adjustments to Line 107-108.

Objectives (104-106): how did you "determine the potential emergence of resistance to copper, zinc, and ceftriaxone based on dietary mineral supplementation commonly found in commercial dairy farms.? Which specific portion of your methods measured the emergency of resistant bacteria? Please justify that whatever method was used is a valid method for that purpose.

Thank you for your comment, but the authors disagree. Culturing is the current state of the art for determination of phenotypic resistance to antimicrobials, as seen in human medicine. Further genomic analysis and annotation of resistance genes was unfortunately beyond the scope of this manuscript. However, to add clarity, the authors have added phenotypic resistance tests to the culturing section title (Line 333).

Reviewer #2 (Comments for the Author):

Thank you for your reply to my comments. In your discussion or introduction, please clearly highlight the effect of extended periods of metals exposure vs. short-term exposure on resistance genes. Animal feed processors may misinterpret your study's conclusions, so it's crucial to emphasize this difference.

Thank you for this comment. We have attempted to address this in lines 68- 72 by quantifying the term short/long exposure relative to dairy cattle.

Please note that authors have addressed my comments, but I still feel that there is a need to reword the findings. I propose rephrasing the conclusions as follows: "At the conditions of our experiments, short exposure to minerals does not affect AMR." If the authors established prolonged exposure to metals, then their statement is incorrect. This is in contrast to the current conclusion: "In conclusion, the over supplementation of minerals had minimal impacts on the gastrointestinal microbiome, indicating that levels of microbial toxicity were not reached..."

Thank you very much for your suggestion. Adjustments have been made to specify the conclusions to our experimental conditions. Please see lines 282-291.

Re: Spectrum01090-24R2 (The influence of trace metal supplementation on the presence of ceftriaxone resistance in Enterobacteriaceae in the gastrointestinal tract of dairy cattle)

Dear Dr. Renee Maxine Petri:

Thank you for appropriately addressing the Reviewer comments, which helped to improve the manuscript. I would hereby like to congratulate you on the acceptance of your paper for publication in Spectrum.

Your manuscript has been accepted, and I am forwarding it to the ASM production staff for publication. Your paper will first be checked to make sure all elements meet the technical requirements. ASM staff will contact you if anything needs to be revised before copyediting and production can begin. Otherwise, you will be notified when your proofs are ready to be viewed.

Sincerely,
Jan Claesen
Editor
Microbiology Spectrum